# Species-Area Relationship and Its Scale-Dependent Effects in Natural Forests of North Eastern China

**Beibei Chen** [1,2]**, Jun Jiang** [1,2,]*[ ] **and Xiuhai Zhao** [1,2]

[1]    Forestry School, Beijing Forestry University, Beijing 100083, China; chenbei@bjfu.edu.cn (B.C.); zhaoxh@bjfu.edu.cn (X.Z.)
[2]    Research Center of Forest Management Engineering of State Forestry and Grassland Administration, Beijing Forestry University, Beijing 100083, China
*    Correspondence: jiang@bjfu.edu.cn

**Abstract:** The Species-area relationship is one of the core issues in community ecology and an important basis for scale transformation of biodiversity. However, the effect of scale on this relationship, together with the selection of an optimal species-area model for different sampling methods, is still controversial. This study is based on the data from two sampling areas of 40 km$^2$ in size, one in a Korean pine (*Pinus koraiensis* Sieb. et Zucc) broad-leaved mixed forest in Mt. Changbai and the other in Jiaohe, Jilin Province. The logarithmic, power, and logistic model were established on a scale of 10 km$^2$, 20 km$^2$, and 30 km$^2$, respectively, using a nested sampling plot and random sampling plot. The goodness of the species-area model was tested by the Akaike information criterion (AIC). The results show that the sampling method affected the relationship between species and area, and the data were fitted better under random sampling compared with nested sampling. The construction of the relationship between species and area was closely related to the upper limit of the sampling area size. On a small scale (10 km$^2$), the data were fitted best with the logarithmic and logistic model, whereas the logistic model was the best fit on a medium (20 km$^2$) and large scale (30 km$^2$). We evaluated the scale dependence of species-area relationship in two forests with nested and random sampling methods. We further showed that the logistic model based on the random sampling plot can explain most soundly the species-area relationship in Jiaohe and Mt. Changbai. More studies are needed in other regions to develop models to optimize sampling designs for different forest types under different density constraints at different spatial scales, and for a more accurate estimation of forest dynamics under long-term observations.

**Keywords:** species-area relationship; sampling; goodness-of-fit; scale-dependent; diversity

## 1. Introduction

A species-area relationship (SAR) describes the rule that the number of species changes when the sampling area increases. This is a basic problem in community ecology and considered as "one of the truly classical theorems in ecology" [1–3]. The species-area relationship can be used to solve many biodiversity problems, such as the estimation of the total number of species on a regional surface [4], evaluation of the loss of regional biodiversity and the diversity of species in hot spots [5–7], and delineation of the optimal area of nature reserves [8]. Therefore, the species-area relationship is an important factor in the conservation and evaluation of regional biodiversity [9] and merits detailed study. The study of the relationship between species and area is affected by different sampling methods, especially area-construction methods [10,11]. The species-area relationship is best constructed with a nested sampling plot; its scale can range from community to region and even to entire continents [12,13]. The earliest method for the construction of the relationship between

species and area is random combined sampling [14], which can be divided into continuous and discontinuous random sampling. Especially after the species pool theory was put forward in the 1990s, random combined sampling has been used more widely, and its research scale has also extended from communities to regions [15].

At different scales, the factors affecting the distribution pattern of biodiversity are different [16]. Therefore, the species-area relationship shows an obvious scale effect, which is mainly reflected in different slopes of the species-area relationship (i.e., the value of power exponent, *z*) at different scales [9–11]. Many studies have used different mathematical models to fit the relationship between species and area; among them, the most commonly used models are the power model [12], the logarithmic model [12], and the logistic model [17]. The three models are simple to apply, and their parameters have clear biological significance, so they are widely accepted [18,19]. Among these models, the power model is the most widely used and has been applied in different sampling methods and at different scales [20].

Although model evaluation and the construction of the species-area relationship have become novel in community ecology, many core issues still need to be solved. In particular, there has been much confusion over the relevance of species–area relationships constructed from sampling-scale data to larger scale biodiversity patterns. For example, how do different sampling methods affect the fitting of the species–area model? It also remains uncertain how to choose the optimal model [21]. Although different expressions of model parameters have generally been accepted, the changes of two parameters, i.e., area and species, as well as their effects on the model, have rarely been reported [22]. In addition, different studies may use different analytical methods to fit the relationship between species and area, and the ecological explanations may also be different. Some studies suggest that with the change of scale (sampling area), the logarithmic model is suitable on a small scale, the power model on a medium scale, and the logistic model on a large scale [23]. Because of the limitation of sampling area size, the relationship between species and area may not be known on a large scale, and the result may be contrary to the widely accepted power relationship [17,24].

As the local climax vegetation, the Korean pine broad-leaved mixed forest is mainly distributed in northeast China and has a complex forest structure and rich biodiversity. Korean pine broad-leaved forest is very important in maintaining the regional biodiversity [25]. However, because of the limitation of research scale and methods, many ecological phenomena and rules, such as the spatial distribution pattern of species, the mechanism of species coexistence, and the internal mechanism of community renewal and succession, have not yet been explained soundly. Moreover, the survey results of small sampling plots may even provide inaccurate information. Especially since the commercial cutting of natural forests was stopped in China in 2017, there have been few reports about the difference between the diversity of species in natural forests after the ban and the diversity of the previous nature reserves.

Using data from very large (40 ha) observational field studies in two representative temperate forests in northeastern China (Figure 1a), we calculated the species–area relationships across all possible quadrats of a given size by using the strictly nested and random sampling method. The objective was to test two sampling methods (nested/random sampling plot) as well as three species-area relationship models (logarithmic, power, and logistic). Our goal is: (1) to investigate the species composition and of the two communities, as well as to provide a basis for clarifying the relationship between species and area, (2) based on the species level of the sampling plot, to use different area sampling methods and different research scales to reveal which sampling method is more suitable to describe the species-area relationship, (3) to explore the scale-dependent effect of the species-area relationship and the implications for the decision-making of forest conservation in this region.

## 2. Materials and Methods

### 2.1. Study Sites

The study was conducted in Northeast China at two sites along a latitudinal gradient from 42°35′ N to 43°57′ N (Figure 1). The annual mean temperature in this region ranges between −2.1 °C and 2.6 °C, and the annual precipitation is between 510 mm and 810 mm. The broad-leaved Korean pine forest, a climax forest of the temperate zone, is characterized by a complex structure and an abundant biodiversity. The study sites covered most of the latitudinal range of the broad-leaved Korean pine forest. The broad-leaved Korean pine forests are the typical temperate mixed forests in northeast China. The dominant tree species are *Pinus koraiensis* Sieb. et Zucc, *Tilia amurensis* Rupr., *Fraxinus mandshurica* Rupr., *Quercus mongolica* Fisch. ex Ledeb., and *Acer mono* Maxim. [26].

The first plot was near a mature broad-leaved Korean pine forest located in the cold temperate zone of Mt. Changbai, Jilin Province (41°23′–42°36′ N, 126°55′–129° E). The broad-leaved Korean pine forest zone in Mt. Changbai is a typical zonal forest with the greatest number of animal and plant species and the most abundant vegetation on this mountain. It mainly grows at an altitude of 720–1100 m in the areas with a mild climate, heavy rainfall, and dark-brown forest soil. The main tree species include *Pinus koraiensis*, *Abies holophylla* Maxim., *Betula platyphylla* Suk., *Carpinus cordata* Bl., *Fraxinus mandshurica*, *Maackia amurensis* Rupr. et Maxim, *Populus ussuriensis* Kom., *Quercus mongolica*, *Tilia amurensis* Rupr., *Ulmus japonica* (Rehd.) Nakai., and *Ulmus laciniata* (Trautv.) Mayr.

The second plot was in a mature broad-leaved Korean pine forest located in Jiaohe, Jilin Province. The study area was located in the forest farm of the Administration Bureau of Jiaohe Forestry Experimental Area, Jilin Province (43°57′–43°58′ N, 127°44′–127°44′ E, altitude: 459–517 m, Figure 1a). The tree species mainly include *Pinus koraiensis*, *Abies holophylla*, *Acer mandshuricum* Maxim., *Acer mono* Maxim., *Betula platyphylla*, *Carpinus cordata*, *Fraxinus mandshurica*, *Juglans mandshurica* Maxim., *Maackia amurensis*, *Quercus mongolica*, *Tilia amurensis*, and *Ulmus laciniata*. The shrubs mainly include *Corylus mandshurica* Maxim., *Euonymus pauciflorus* Maxim. and *Rhamnus davurica* Pall.

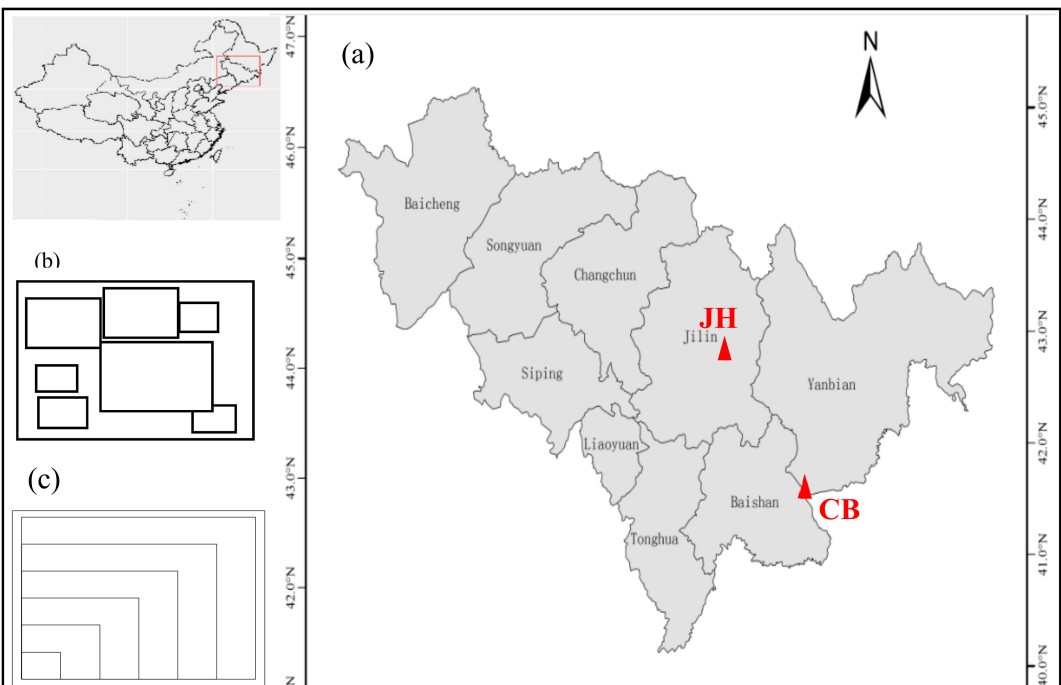

**Figure 1.** (**a**) The sampling plots (red triangles) of the broad-leaved *Pinus koraiensis* forests in Yanbian City in Jilin Province (CB), and the secondary coniferous and broad-leaved mixed forest in Eastern Jiaohe City in Jilin Province (JH), China, (**b**) and (**c**) random and nested sampling plots, respectively.

## 2.2. Sampling Plot Setting and Data Survey

At the study sites described above, fixed sampling plots of 40 km$^2$ in size were established in 2015. Both plots were composed of *Pinus koraiensis* and broad-leaved species such as *Acer* spp., *Fraxinus mandschurica*, and *Tilia* spp. Both plots were divided into 1050 continuous sampling plots each of which was 20 × 20 m in size and the continuous sampling plots were further divided into 16 subsampling plots each of which was 5 × 5 m in size. In each plot, the name of an individual species, diameter at breast height (DBH), tree height, under-branch height, and crown width were identified, measured, and mapped for each plant with DBH > 1 cm [26,27]. The overview of the sampling plots is shown in Table 1.

**Table 1.** A summary of permanent forest plots in Mt. Changbai and Jiaohe.

| Site | Forest Type | Average Elevation (m) | Total Basal Area (m$^2$/km$^2$) | Dominant Species |
|---|---|---|---|---|
| Mt.Changbai | Primary *Pinus koraiensis-Tilia amurensis* mixed forests | 779 | 25.36 | *Pinus koraiensis* *Fraxinus mandshurica* *Tilia amurensis* *Quercus mongolica* |
| Jiaohe | Secondary coniferous and broad-leaved mixed forests | 517 | 21.28 | *Pinus koraiensis* *Carpinus cordata* *Juglans mandshurica* *Fraxinus mandshurica* |

## 2.3. Sampling Methods

The species-area relationship can be constructed in three main ways, namely with a nested sampling plot, and a random sampling plot [24]. In this study, we employed the nested and random sampling plots. For the bottom left nested sampling plot in the given sampling plot (840 × 500 m), a total of 32 sampling areas which were 5 × 5 m, 10 × 10 m, … 500 × 500 m, and 840 × 500 m in size were selected in turn, in a way that larger sampling areas enclosed smaller sampling areas. The number of species falling in the corresponding sampling areas was calculated using software R as the number of species falling in the current side length of the sampling area (Figure 1b). The random sampling plot refers to the discontinuous combined sampling method revised by Scheiner [24]. In a given sampling plot (840 × 500 m), a total of 32 sampling areas which were 5 × 5 m, 10 × 10 m, … 500 × 500 m, and 840 × 500 m in size were selected randomly and conducted with the software R, and then the number of species falling in each sampling area was counted (Figure 1c). To reduce the number of missing species caused by random sampling, the average number of species of 1000 times random samplings was calculated as the number of species in each sampling area.

## 2.4. The Evaluation of the Goodness-of-Fit of Species-Area Models

The formula of the models used in this study are as follows:
Logarithmic model [14],

$$S = c + z\text{Log}(A) \tag{1}$$

power model [12],

$$S = cA^z \tag{2}$$

and logistic model [28],

$$S = c/(1 + \exp(-z\,A + b) \tag{3}$$

where $S$ is the number of species, A is the sampling area, and $c$, $b$, and $z$ are constants [12,24].

The statistical analysis and mapping in this study were completed using the R 3.5.1 statistical software and Excel 2010. The logarithmic and the linear model were tested with the method of

least squares, and the fitting of the power and logistic model was tested using the Gauss–Newton algorithm [29]. Moreover, the Akaike information criterion (AIC) was used to evaluate the goodness-of-fit of the three models under different sampling methods [30].

The formula is as follows:

$$AIC = -2L + 2K \tag{4}$$

where $L$ is the logarithmic likelihood value of the model, and $K$ is the number of parameters of the model. The smaller the AIC value, the better the model fitting and the more accurate the predicted value [30].

## 3. Results

### 3.1. The Composition of Species and Structural Characteristics of Dominant Populations

In Mt. Changbai, 74,652 individual woody plants belonging to 54 species, 32 genera, and 19 families were detected in the sampling plot. The most abundant genera were *Aceraceae* with eight species, followed by *Caprifoliaceae* with five species and *Sapindaceae* with four species. A total of 54,652 individual woody plants belonging to 50 species, 29 genera, and 19 families were found in the 40 km$^2$ plot in Jiaohe. According to the plant genus floristic classification criteria [31], the main distribution was the North temperate zone distribution type including 23 genera, which accounted for 71.96% of all genera in Jiaohe (Table 2). The families with the largest number of species were *Aceraceae* with seven species, followed by *Caprifoliaceae* with three species and *Sapindaceae* with three species. There were 15 single genera, which accounted for 53.6% of all genera.

**Table 2.** Areal types of woody plants in Mt.Changbai and Jiaohe plots.

| Site | Mt. Changbai | | Jiaohe | |
|---|---|---|---|---|
| Distribution Area Type | No. of Genera | Proportion (%) | No. of Genera | Proportion (%) |
| Cosmopolitan distribution | 1 | 3.12 | 1 | 3.44 |
| Pantropic distribution | 1 | 3.12 | 1 | 3.44 |
| North temperate distribution | 23 | 71.96 | 21 | 72.41 |
| Old World temperate distribution | 1 | 3.12 | 1 | 3.44 |
| East Asian distribution | 5 | 15.62 | 4 | 13.79 |
| Sino-Japanese distribution | 1 | 3.12 | 1 | 3.44 |
| Total | 32 | — | 29 | — |

### 3.2. The Species-Area Relationship Obtained by Nested Sampling

The results showed that the logistic model was the best among the three models, with the AIC values of 74.62 in Mt. Changbai and 84.67 in Jiaohe, followed by the power model (Table 3). The values predicted by the logarithmic model differed significantly from the observed values; the number of predicted species was underestimated, and fitting was the least accurate among the three models.

**Table 3.** The evaluation of the goodness-of-fit of the three species-area models using nested and random sampling designs.

| Sampling Design | Model | Mt. Changbai | | | | Jiaohe | | | |
|---|---|---|---|---|---|---|---|---|---|
| | | c | z | A | AIC | c | z | A | AIC |
| ND | Logarithmic | −18.36 ** | 7.33 ** | — | 108.11 | −22.16 ** | 5.46 *** | — | 129.97 |
| | Power | 2.18 *** | 0.18 *** | — | 97.23 | 3.11 *** | 0.22 *** | — | 95.91 |
| | Logistic | 1.99 ** | 0.24 *** | 0.03 *** | 74.62 | 1.75 | 0.32 *** | 0.02 *** | 84.67 |
| RD | Logarithmic | −14.32 ** | 4.31 *** | — | 97.23 | −20.43 *** | 5.52 *** | — | 89.11 |
| | Power | 3.98 *** | 0.23 *** | — | 78.24 | 4.46 *** | 0.19 *** | — | 113.81 |
| | Logistic | 1.25 *** | 0.28 *** | 0.04 *** | 69.33 | 1.45 *** | 0.39 *** | 0.02 *** | 71.21 |

ND: nested design, RD: random design, AIC: Akaike information criterion, ** $p < 0.01$, *** $p < 0.001$. c, z are constants, A is the sampling area.

### 3.3. The Species-Area Relationship Obtained by Random Sampling

The results show that under random sampling, the best AIC values of the logistic model were 69.33 in Mt. Changbai and 71.21 in Jiaohe (Table 3). The values predicted by the power and the logarithmic model obviously differed from the observed values, and the fitting results of all three models were relatively poor. Comparing the nested and random sampling, we can see that the species–area relationship yielded by the power model under nested sampling was relatively good, whereas, the species–area relationship was best described by the logistic model under random sampling in both studied forest communities.

For both forest types, different sampling methods had significant effects on the species-area relationship in the largest sampling plots (42 km$^2$). The random sampling plot was a better fit for the data than the nested sampling plot. The results suggested that the power function provided a consistently better fit to fine-scale richness data, as revealed by the consistently lower AIC values. The AIC values of the logistic model ranged from 50 to 90 (Figure 2, $p < 0.001$). Fattorini [32] showed that the sampling method could affect the construction of the species-area model.

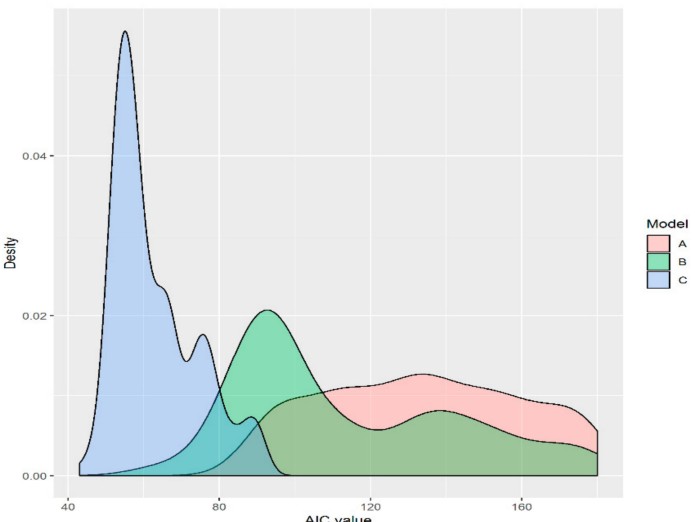

**Figure 2.** The density distribution of the Akaike information criterion values for three models fitted to the species–area data ($p < 0.001$). A: logarithmic model, B: power model, C: logistic model.

### 3.4. The Species-Area Relationship at Different Scales

In line with the recommendations by He and Legendre [17] for different scales and the minimum area of the whole sampling plot, three scales were selected based on the size of the sampling plot in this study. These included a small scale with the upper size limit of 10 km$^2$, a medium scale with the upper size limit of 25 km$^2$, and the whole 42 km$^2$ sampling plot. The goodness-of-fit of the three species–area models was tested (Table 4), and the effect of the upper limit of the sampling area on the species-area relationship was studied. Moreover, the goodness of three species–area models at different scales or area limits was compared. The results show that the AIC values for the power model were the smallest on a small scale, with the values of 86.62 for Mt. Changbai and 81.11 for Jiaohe. That is, the power model was the best fit for the data. The AIC values of the power and logistic models were similar on a medium scale, with the power model as relatively optimal. On a scale of 40 km$^2$, the best species–area model was the logistic, with the AIC values of 56.63 for Mt. Changbai and 61.28 for Jiaohe.

**Table 4.** The evaluation of the Akaike information criterion for the three species-area models of varying size.

| Sampling Design | Model | Mt. Changbai | | | Jiaohe | | |
|---|---|---|---|---|---|---|---|
| | | 10 km$^2$ | 25 km$^2$ | 40 km$^2$ | 10 km$^2$ | 25 km$^2$ | 40 km$^2$ |
| ND | Logarithmic | 117.23 | 158.21 | 106.38 | 121.93 | 144.61 | 113.22 |
| | Power | 86.62 | 100.23 | 97.36 | 81.11 | 97.62 | 95.46 |
| | Logistic | 88.56 | 98.66 | 97.26 | 82.55 | 94.44 | 83.23 |
| RD | Logarithmic | 114.26 | 147.89 | 100.59 | 98.34 | 123.55 | 111.97 |
| | Power | 65.89 | 90.22 | 96.31 | 62.57 | 64.57 | 116.32 |
| | Logistic | 90.26 | 88.62 | 58.63 | 97.42 | 99.76 | 61.28 |

ND: nested design, RD: random design.

## 4. Discussion

### 4.1. The Optimal Species-Area Model

In a random sampling plot, an area of a predefined size is set up randomly, and the number of species in the sampling plot is counted so that all parts of the community have the same opportunity to be selected. In this way, a series of scattered samples is generated, which may randomly encounter more species and can reduce the effect of the distribution of species clusters. Thus, random sampling plot is the best sampling method for the species–area model. Constructing the species-area relationship using a combinatorial sample series in multiple regions can effectively estimate the biodiversity of large regions and provide a new method to assess the community and regional species pools [33]. Nested sampling strictly extracts small samples from the overall sample by setting a large sampling plot enclosing several smaller sampling plots with uniform distribution. However, rare species in the smaller sampling plots easily appear in the large sampling plot, resulting in an overestimation of the species number and reducing the prediction accuracy. Gleason et al. [34] also showed that random sampling is better than nested sampling because the latter may be affected to some extent by species clustering and environmental heterogeneity. In this study, the results showed that the species-area relationship in the studied community could be well fitted by both the power and the logistic models. However, the logistic model was slightly better than the power model and could also accurately fit the species-area relationship on a large scale.

### 4.2. Scale Dependence of the Species-Area Relationship

The species-area relationship depends on the scale, and the goodness-of-fit of the model on different scales must be considered when selecting the model. Fridley et al. [9] found that the logarithmic model is suitable for species-area relationship on a small scale, the power model is suitable for a medium scale, whereas the logistic model is most suitable for a larger scale [17]. However, Ulrich and Buszko [35] believe that the power model might show an infinite growth on a small scale, leading to a serious overestimation of the number of species in the sampling plot. They also showed that the overestimation of the number of species by the power model is not obvious on small and medium scales, and it may cause prediction errors if applied to large-scale research areas.

He and Legendre [17] pointed out that the logistic model can soundly describe the species-area relationship on a larger scale. As long as the sampling area is large enough to cover all species in the region, the logistic model is better than other models. When the AIC test is used to select the optimal model, the logistic model combined with random sampling is better than the power model for large-scale research areas. This may be directly related to the division of research scale and the number of species covered by the sampling area. In this study, the sampling plot covered all the species in the studied forests, and the extracted area was large enough that the logistic model could accurately

describe the species-area relationship. This result for Mt. Changbai and Jiaohe using the logistic model are consistent with those of a study by He and Legendre [17].

The factors affecting the distribution pattern of biodiversity at different scales are different. Therefore, the relationship between species and area shows obvious scale effect, which is mainly reflected in the difference of slope (i.e., *z* value) of the relationship between species and area on different scales. With an increase in the spatial scale, the probability of rare species loss decreases, and the variation among different samples decreases, showing a relatively stable species–area curve. In addition, the change in the species–area curve with scale might also be related to the difference in habitat types. White et al. [36] found that continental species-area relationships consist of three major sub-patterns (triphasic) at different scales, and the coordinate axis is "S" shape. Our findings agree with those of White et al. [36] with respect to the triphasic species–area relationship (Figure 3), where there is a rapid increase in species with area as a consequence of individual sampling, then processes generating fractal like distributions of species operate to produce power laws with slopes. The species-area relationship at the local scale is influenced by sampling effect and stochastic process, and the slope is larger. On a large scale, the difference in habitat types as well as in communities was small among the samples, thereby leading to a stable species–area curve.

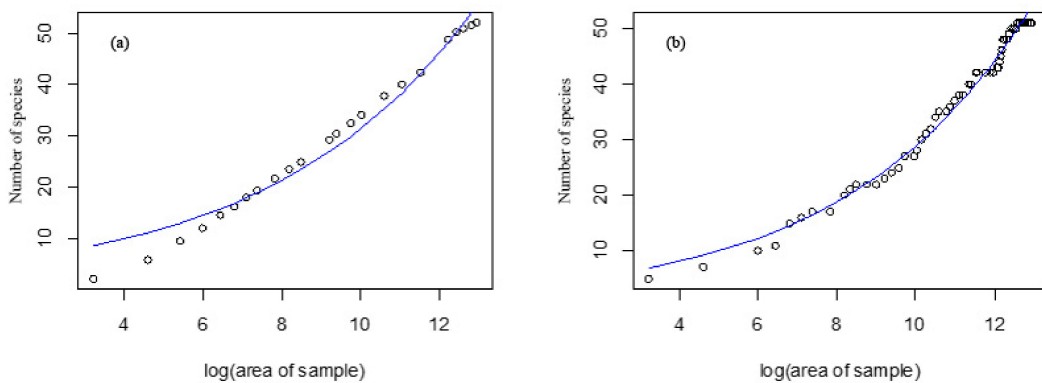

**Figure 3.** Log-log species-area curve of the logistic model of Jiaohe (**a**) and Mt. Changbai (**b**).

*4.3. The Effect of Habitat Heterogeneity on Species-Area Relationship*

The species-area relationship also reflects to some extent the spatial distribution characteristics of the community [37]. Larger areas usually have more habitats and greater heterogeneity, both of which are generally positively correlated with species richness [38]. Our finding also supports this claim. Because of the natural sparseness, disturbance pattern, and environmental changes of a forest, the sampling plot of coniferous and broad-leaved mixed forests changed the environmental conditions of light and soil after forest disturbance, increased the heterogeneity of community habitat, enhanced the function of habitat filtration, and enabled coexisting species to utilize similar resources, as well as the degree of community aggregation increased significantly and the number of species increased rapidly with the increase of area. With an increase in the sampling area, the change in the habitat heterogeneity was weakened, the emergence of new species was gradually reduced, and the number of species decreased until it became stable. This also explains why the species at the end of the species–area curve mostly belong to the rare or occasional species in the community, and why the change of the curve might be affected by random processes such as patch dynamics and diffusion restriction. These species were small in number and scattered in distribution, but their existence might have been supported by the tradeoff in their adaptability, such as strong stress tolerance and slow growth rate. Some species even completely stop growing and wait dormant for suitable living conditions to emerge [39].

Many studies have introduced and defended the power species–area model primarily on the basis of its more reasonable approximation of richness at much larger scales [40]. However, this is not a result

of the logistic function being a poor fit to fine-scale data, but rather that log-log species-area curves are triphasic, with slower accumulation rates in an intermediate area. The species-area relationship at the local scale is influenced by sampling effect and stochastic process, and the slope is larger. The species-area relationship at the regional scale is mainly affected by habitat heterogeneity, with a smaller slope. At the regional, intercontinental and global scales, the slope is larger because it contains different biogeographic regions and different seed banks. Our results highlight the consistency of scale-dependent effects for the species-area relationship and suggest consistency with larger-scale biogeographic patterns. Fridley et al. [9] found rates of species accumulation (z values) at fine scales to be consistently higher than those generally reported for intermediate scales, this implies some statistical or biological constraint over the spatial distribution of species richness at small scales. Whereas, Kerr and Parler [41] showed that species diversity is more dependent on habitat heterogeneity, and the impact of area or habitat diversity on species diversity is more significant than that in severe climatic regions. In addition, richness should be fundamentally constrained by the density of individuals in small quadrats, and therefore estimating individual density for species is notoriously difficult. Even the backward species–area relationships is not the way to estimate extinction rates from habitat loss [42,43].These results indicate that the species-area relationship might have a recognizable pattern with separation of sampling, and further research is needed to understand the sampling and measurement properties of the different types of species-area relationships.

## 5. Conclusions

The results of this study show that the logistic model based on the random sampling plot can explain most soundly the species-area relationship in Jiaohe and Mt. Changbai. It should be noted that the species-area relationship studied is only valid for the typical communities in the region because this relationship is also affected by community succession and spatial distribution. Whether the regional species-area relationship is suitable for other forest communities remains to be further confirmed. In this study, the coniferous and the broad-leaved mixed forest community in Jiaohe is in the stage of recession of pioneer tree species, where several dominant tree species such as *Pinus koraiensis* and *Acer mono* coexist. Because of the habitat changes caused by community succession after the disturbance, increasing the sampling area leads to the detection of new species and habitat types [44]. In addition, increasing the sampling area also leads to increased species mobility and decreased extinction rate, which jointly cause an apparent increase in species richness in the sampling area [45]. As a result, the species-area relationship differs from the previous succession stage. Therefore, the selection of the species–area model should consider not only the statistical significance of the model but also be interpreted and selected according to the actual situation and ecological theory.

However, in the process of field investigation and extrapolation calculation, attention should be paid to the following points. Firstly, field investigation should cover all vegetation types in the region as widely as possible. Secondly, the parameters of the species-area relationship in each region should be specific, identification of the principal biological or statistical constraints on species accumulation rates at different spatial scales, including the density constraints at various scales. Thirdly, to help strengthen the ecological explanation behind forest conservation planning, we need far better data and long-term observations to find a balance between the mathematical model itself and its practical application.

**Author Contributions:** Conceptualization, J.J.; data curation, J.J. and B.C.; formal analysis, J.J.; funding acquisition, J.J. and X.Z.; investigation, J.J. and B.C.; methodology, J.J. and X.Z.; writing—original draft, J.J. and B.C.; writing—review and editing: B.C. and J.J.

**Funding:** This work was funded by the Fundamental Research Funds for the Central Universities (2018ZY27) from the Beijing Forestry University and the National Key Research and Development Program of China (2017YFC0504005).

**Acknowledgments:** We thank two anonymous reviewers for helpful comments on earlier versions of this manuscript.

**Conflicts of Interest:** The authors declared no potential conflicts of interest with respect to the research, authorship, and/or publication of this article.

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
