# Peer review of "Species-Area Relationship and Its Scale-Dependent Effects in Natural Forests of North Eastern China"

_forests, doi:10.3390/f10050422_

Round 1
Reviewer 1 Report
This is generally an informative, easy to follow article with clear implications to ecological research. The biggest issue is what is going on with Figures 2 (line 184 comments) and 3 (lines 198, 200 and 223 comments). A few times in abstract and discussion/conclusion sections, the article seems to extrapolate the utility of the article without making clear logical connections to why the utility may extend to these concepts. Otherwise, this is a good article to read that extends the field.
Line 23. Not clear why this is included, since its relation to the other AIC values is not shared.
26. It is not clear, from just the abstract, how the results show that attention should also be paid to habitat change and community succession, since those are not previously mentioned.
32. It is not clear how the previous statement leads into this statement ("implies that ecological processes...") More connection needs made for this transition.
37. It would be useful to the reader to mention by name these processes that only happen at large scales
38 (also 41 and multiple others). Generally, I prefer to read "The species-area relationship" or "SAR", not as written.
54. No citation for power model? At the very least, there should be a comma between it and log model.
59-60. "hot topic" does not seem an appropriate term here. "novel" or "fashionable" would likely work better
73-74. What is meant by "unique population"? Also, a citation describing the flora of the broad-leaved Korean pine forest would be useful to a non-Chinese reader.
82. Unclear what is meant by the acronym "BCI"
84. Could include reference to Fig. 1(a) here)
104. Period missing.
113. "Mature"
134. "Forest Type" and "Primary Vegetation" are very redundant here. I would also be curious to know how many stems per sampling plot and how many species per sampling plot were in these two sites.
145. How were the samples selected randomly? I assume through a software package?
150-151 (and 156). Useful to separate equations out onto their own lines, so that it is easier for the reader to interpret.
152. While I personally understand the models, the specifications of c, b, and z should be explained in further detail than "model parameters."
153-154. Citations could be included to support these methods of testing.
157. Better to read "Smaller AIC values correspond to a better model fit and more accurate predicted value."
159. Put this up at the beginning of this section.
163. Acer and Lonicera are not families, they are genera. (Sapindaceae and Caprifoliaceae, respectively, are their families).
173. How many species for these two genera?
162-173. Lots of numbers. Most of this information would be easier to digest in a table.
184. While the AIC generally indicates logistic model is best, the eye test seems that logistic does not accurately capture the nature of the data in Jiaohe. In Mt. Changbai I can accept its utility, but the model and the data seem to be in general disagreement in Jiaohe.
198. Do you mean Fig 3?
200. Y axis hard to read. Doesn't this indicate that C (Blue) is the best fit, which is labeled as logarithmic?
219. Seems to be a sentence fragment.
223. This does not seem to be what Fig. 3 is showing (or what the Fig. 3 caption is describing...)
224. Awkward phrasing. Is there a missing period?
251. Try to avoid "obviously", since it may not be obvious to all readers.
270. "Ecological drift" needs to be clearly defined here.
281. "Believes" is an awkward term for a publication 38 years old.
307. Extra space after Willig
366. Name formatting is different from others.
371-2. Spacing and "U.S." errors
Author Response
Dear reviewers,
Thank you very much for your letter and advice. We would like to upload the enclosed entitled “Species-area relationship and its scale-dependent effects in natural forests of North Eastern China”We have revised the paper, and would like to re-upload it for your consideration. We have studied reviewer’s comments carefully and have made revision which marked, using the Track Changes function in Microsoft Word in the paper, and the amendments are highlighted in red in the revised manuscript. In the attachment you will find our point-by-point responses to the reviewers’ comments/ questions.
According to the reviewer’s comments, we have revised the manuscript extensively, we also revised some of the discussions in detail, shorted and keep context with results,and revised the places related to the results and some conclusions, add some interpretation on last section of discussion, which mentioned by the reviewers. Additionally, the participants in this revision also made great contributions, so we added and adjusted the author's order, and the author(s) declared no potential conflicts of interest with respect to the research, authorship, and/or publication of this article.
We hope that these revisions are satisfactory and that the revised version will be acceptable for publication. We would like to express our great appreciation to you and reviewers for comments on our paper.
Looking forward to hearing from you.
Best wishes,
Yours sincerely,
Jun Jiang

Reviewer 2 Report
The review of the manuscript forests 495742
Title: Species-area relationship and its scale-dependent effects in natural forests of Northeast China
Authors: Jun Jiang *, Beibei Chen
The paper is focussed on important topic of Species-area relationship and its scale-dependent effects in Korean pine forests and this topic is interesting for an international reader. Aims or questions should be defined clearly, and all other chapters should follow these aims. Data is well-statistically processed. Chapter Materials and Methods, Results it should be explained more detail.
The weakness of the article is unclearly explained some results and some parts of results are confusing (The species-area relationship at different scales). I propose to markedly add some interpretation. It is important for the reader to make it clear. Similarly, some conclusions are not confirmed by results in chapter Discussion. I propose to short the discussion and to keep context with yours results. The citations have to be unified.
Other comments:
Line:
100 – “ four sites” Why four? This study interpreted results from only two sites (plots)! Please correct.
106 – Expression Broad-leaved Korean pine forests – is confusing Broad-leaved and pine? Maybe better Broad-leaved with Korean pine forests or otherwise
137- Why do you mention a method you didn't use in this study? “Isolated sampling method” please remove
174 – Table 2 – please all number in the same decimal
183 – You explain, why is the number of cases in left and right graph (in figure 2) is different? I don’t understand, nothing was explained in methods …
198, 204, 305, 312 – citations - sometimes with year of publication, sometimes not…please unify
216-218 – you interpreted some minimum sampling area, but no specific number is mentioned in the results - what was the minimum sampling area- please add specific number,
This part sounds like a discussion and explanation is no so good, these two plots are different in composition - please change or add more exactly explanation
219-228 – this part of the results is very confusing, figure 3 is about the density of the Aikaki information criterion values and you explain scale effect, z values in whole part – please reworded complete the whole paragraph
298-316 –This part is not directly related to the results, please shorted and keep context with results

Author Response

(The authors gave the same response as above.)
